# Characterisation of Cell-Mediated Immunity Against Bovine Alphaherpesvirus 1 (*BoAHV-1*) in Calves

**DOI:** 10.3390/vaccines13100996

**Published:** 2025-09-23

**Authors:** Giulia Franzoni, Cecilia Righi, Immacolata De Donato, Giovanna Cappelli, Giovanna De Matteis, Eleonora Scoccia, Giulia Costantino, Emanuela Giaconi, Susanna Zinellu, Carlo Grassi, Alessandra Martucciello, Francesco Grandoni, Stefano Petrini

**Affiliations:** 1Department of Animal Health, Istituto Zooprofilattico Sperimentale della Sardegna, 07100 Sassari, Italy; emanuela.giaconi@izs-sardegna.it (E.G.); susanna.zinellu@izs-sardegna.it (S.Z.); 2National Reference Centre for Infectious Bovine Rhinotracheitis (IBR), Istituto Zooprofilattico Sperimentale Umbria-Marche, “Togo Rosati”, 06126 Perugia, Italy; e.scoccia@izsum.it (E.S.); g.costantino@izsum.it (G.C.); s.petrini@izsum.it (S.P.); 3National Reference Centre for Hygiene and Technology of Breeding and Buffalo Production, Istituto Zooprofilattico Sperimentale del Mezzogiorno, 84131 Salerno, Italy; immacolata.dedonato@izsmportici.it (I.D.D.); giovanna.cappelli@izsmportici.it (G.C.); grassicarlo@libero.it (C.G.); alessandra.martucciello@izsmportici.it (A.M.); 4CREA-Consiglio per la Ricerca in Agricoltura e l’Analisi dell’Economia Agraria, Centro di ricerca Zootecnia e Acquacoltura (Research Centre for Animal Production and Aquaculture), 00015 Monterotondo (RM), Italy; giovanna.dematteis@crea.gov.it (G.D.M.); francesco.grandoni@crea.gov.it (F.G.)

**Keywords:** BoAHV-1, T cells, monocytes, flow cytometry, cytokines

## Abstract

**Background:** Bovine alphaherpesvirus 1 (*BoAHV-1*) is a major respiratory and reproductive pathogen in cattle worldwide. Both innate and adaptive immune responses contribute to protection against this virus; however, virus-host interactions remain partly undefined. In this study, the impact of *BoAHV-1* infection on calves’ immune responses was investigated in detail. **Methods:** Six calves were intranasally infected with wild-type *BoAHV-1*, and blood samples were collected longitudinally. Leukocyte subset dynamics were assessed by complete haematological assay and flow cytometry, while multiplex ELISA was used to quantify serum levels of ten cytokines. For each parameter, post-infection values (days 2, 4, 8, 10, and 14) were compared with pre-infection baseline values (day 0). **Results:** Infection induced an initial phase of immunosuppression, reflected by decreased circulating αβ and γδ-T cells. However, infected animals rapidly developed a protective immune response, characterised by increased circulating classical and intermediate monocytes and elevated levels of the related chemokine MIP-1β. Early post-infection, rises in serum IFN-γ and IL-10 were also detected. **Conclusions:** Our data suggest that monocyte recruitment and increased serum levels of IFN-γ and IL-10 are positively associated with the ability to overcome infection. A better understanding of the immunopathogenic mechanisms underlying *BoAHV-1* infection will support the development of more effective vaccines against this virus.

## 1. Introduction

Bovine Alphaherpesvirus 1 (*BoAHV-1*) is a DNA virus belonging to the genus *Varicellovirus* in the *Herpesviridae* family, subfamily *Alphaherpesvirinae* [1]. The virus is responsible for Infectious Bovine Rhinotracheitis (IBR), Infectious Pustular Vulvovaginitis (IPV) and Infectious Pustular Balanoposthitis (IPB). Also, the etiological agent is involved in conjunctivitis, necrotic rhinitis, abortion, epididymitis, and metritis, with subsequent infertility problems [2]. The virus poses a significant threat to the global cattle industry due to the substantial economic losses. These include the loss of replacement heifers, reduced milk production, anoestrus and repeat breeding, culling of mature animals, and high veterinary costs [2,3]. In addition, infected cattle often exhibit lymphocytopenia, which subsequently impairs the host immune response, thereby facilitating the replication of other respiratory pathogens. This immunosuppression state contributes to the development of the polymicrobial disorder known as bovine respiratory disease complex (BRDC) [4]. Often, cattle can survive infection and develop an adaptive immune response, which can clear the virus but usually leads to *BoAHV-1* latency, mainly in sensory neurons [4]. In endemic countries, vaccination is often applied as a control measure. Diverse types of products are available, including modified-live (MLV) and inactivated vaccines. It is also possible to differentiate between infected and vaccinated animals using the “IBR *marker* vaccines,” which are authorised in parts of the European Union participating in IBR control or eradication programmes [5,6]. Unfortunately, current vaccines are unable to prevent virus excretion and virus latency in challenged animals [5,6,7]. A better understanding of *BoHV-1’s* impact on the calves’ immune system would benefit the development of safe and effective prophylactic measures against this virus.

After an initial transient phase of immunosuppression, cattle recover from the disease. This recovery is attributed to the development of both innate and adaptive immune responses [4]. Regarding adaptive immunity, both humoral and cellular responses contribute to protection against *BoAHV-1*. Cellular immune responses include both T helper (Th)1 and Th2 cells [8], the Th1 response being primarily involved in the control of infection [9,10]. Thus, several *BoAHV-1* vaccines are designed to trigger both branches of the adaptive immune system.

Previous studies measured Th1 response using ELISA, ELISPOT, or proliferation assays [10,11,12]. Researchers observed that IFN-γ-releasing cells correlated with decreased clinical symptoms [11,12]. Nevertheless, few studies have utilised flow cytometry to investigate leukocyte population dynamics during BoAHV-1 infection [13,14].

In addition, current knowledge of cytokine profiles during *BoAHV-1* infection remains incomplete. Previous studies reported that infection with *BoAHV-1* leads to dysregulation of the usual cytokine profile in vivo. Specifically, several cytokines are released in response to infection, directly or indirectly inhibiting virus replication by activation of immune effector cells [8,15,16]. A Th1 response generally correlates strongly with protection, with elevated circulating levels of interferon-gamma (IFN-γ) serving as a clear marker [10]. Additionally, pro-inflammatory cytokines are secreted early in infection to help suppress viral replication [8,15,17]. In this conceptual framework, contrasting results have been reported in the literature, and some cytokines have not been investigated [10].

The impact of *BoAHV-1* on the immune system remains only partially defined, and we believe that researchers have yet to fully utilise the contemporary veterinary immunological toolbox to characterise the immunopathogenic mechanisms underlying *BoAHV-1* infection.

In this work, we aimed to gain a better understanding of the cellular immune responses to *BoAHV-1* infection. We employed an integrated approach involving a complete haematological assay, four multicolour flow cytometry panels, and a multiplex ELISA for cytokine profiling. The data generated in this study enabled us to better depict crucial host-virus interactions and will hopefully aid in the design of safe and effective control strategies for *BoAHV-1* infections.

## 2. Materials and Methods

### 2.1. Ethical Statement

The in vivo experiments were conducted at the Istituto Zooprofilattico Sperimentale of Mezzogiorno (IZS ME), Salerno, Italy, in accordance with European legislation on the protection of animals used for scientific purposes. The experimental protocol for the care, handling, and sampling of animals was approved by the Italian Ministry of Health (Authorisation number 203/2021-PR) and by the IZS ME Review Board responsible for animal welfare (Organismo Preposto al Benessere e Cura degli Animali) (opinion number of the approved project: 01/2021; date 28 January 2021).

### 2.2. Animal Experiments

Six healthy Italian Friesian calves, aged 3–6 months, seronegative to *BoAHV-1,* were selected for the study. The animals were transferred from the farm to the experimental facility of IZS ME and acclimatised for seven days prior to the start of the experiment. Subsequently, the calves were intranasally challenged with the wild-type infection (wt) *BoAHV-1* strain 16453/07 TN. Clinical scores and rectal temperatures were monitored throughout the study, as previously described [18]. Blood samples were collected at 0, 2, 4, 8, 10, and 14 days post-infection (dpi). A schematic representation of the study design is presented in Figure 1.

### 2.3. Collection of Blood Samples

Approximately 16 mL of whole blood was collected from each animal; K3-EDTA tubes (Vacuette^®^, Greiner Bio-One, Cassina de Pecchi, Italy) were used for haematological analysis, while Li-Heparin tubes (Vacuette^®^, Greiner Bio-One, Cassina de Pecchi, Italy) were used for flow cytometry (see Section 2.4). Additionally, whole blood collected in tubes containing a clot activator (Vacutest^®^, Vacutest Kima s.r.l., Padua, Italy) was used for serum extraction to monitor changes in various serum cytokine levels by multiplex ELISA (see Section 2.5); these serum samples were stored at −80 °C until analysis.

### 2.4. Haematological and Flow Cytometry Analysis

The total leukocyte count (WBC) and main leukocyte subpopulations were evaluated using a Cell-Dyn 3700 SL haematology analyser (Abbott, IL, USA) in EDTA blood, according to the manufacturer’s instructions. Fresh heparinised whole blood samples (50 μL) were analysed using flow cytometry (CytoFLEX flow cytometer, Beckman Coulter, Brea, CA, USA). In detail, three multicolour flow cytometric panels were designed to identify different subsets of lymphocytes and monocytes (Table 1). Panel 1 evaluated the percentage of total (CD3^+^), helper (CD4^+^), cytotoxic (CD8^+^), and γδ-T lymphocytes, as well as their respective subsets. Panel 2 assessed the percentage of total B lymphocytes (CD79^+^) and their subsets (CD21^+^ and CD21^−^). Finally, Panel 3 evaluated the percentage of total, classical (cM), intermediate (intM), and non-classical (ncM) monocytes [19]. For each panel, a specific gating strategy was developed, as shown in Figure 2, Appendix A. A preliminary analysis assessed leukocyte viability (95–98%) using the Live/Dead^®^ Kit (Thermo Fisher Scientific, Waltham, MA, USA) and examined their distribution in the FSC-A vs. SSC-A dot plot. The results were used to establish the “Debris Excluded” gate, which enabled the removal of most cellular debris and non-viable (dead or dying) cells (Figure 2D, Appendix A). Fluorescence minus one (FMO) controls were used to set the quadrant gates shown in Figure 2G,H, as well as in Appendix A, ensuring accurate discrimination of positive and negative populations. For Panels 1 and 3, each blood sample was incubated with 12 µL of antibody mixture for 20 min at 4 °C in the dark. Then, erythrocytes were lysed using 1 mL of TRIS-buffered ammonium chloride solution (0.87% *w*/*v*, pH 7.3) for 10 min at room temperature (RT). After the addition of 4.0 mL of cold PBS, cells were centrifuged at 300× *g* for 5 min and then resuspended in 120 μL of PBS until acquisition. For Panel 2, due to the intracellular localisation of CD79a, the HM47 clone was used with the PerFix-NC Kit (Beckman Coulter), following the manufacturer’s instructions. All labelled samples were immediately acquired using a CytoFLEX flow cytometer, and the data were analysed with CytExpert v2.4 software (Beckman Coulter). The absolute counts for each subpopulation were determined using a dual-platform approach. This involved multiplying their flow cytometrically assessed percentages within the total leukocytes (CD18^+^ cells) by the absolute WBC count obtained from a haematology cell analyser.

### 2.5. Evaluation of Serum Cytokines Levels

Serum samples were collected during the in vivo animal experiment. Whole blood without anticoagulant was centrifuged at 850× *g* for 10 min; the serum was collected and stored at −80 °C until analysis. Levels of IFN-γ, IL-1α, IL-1β, IL-4, IL-6, CXCL8, IL-10, IL-36Ra, MIP-1β, and TNF-α were monitored at 0, 2, 4, 8, 10, and 14 days post-infection (dpi) using the Bovine Cytokine/Chemokine Magnetic Bead Panel Multiplex assay (Merck Millipore, Darmstadt, Germany) and a Bio-plex MAGPIX Multiplex Reader (Bio-Rad, Hercules, CA, USA), following the manufacturer’s instructions [20]. All samples were tested in duplicate (two technical replicates).

### 2.6. Statistical Analysis

The Shapiro–Wilk test was used to determine whether the data for each independent variable followed a normal distribution. Subsequently, differences between values post-infection (2, 4, 8, 10, 14 dpi) and values pre-infection (day 0) were investigated using the parametric Student’s paired *t*-test or the Wilcoxon Mann–Whitney non-parametric test. The Bonferroni correction was used to control bias due to multiple comparisons (*p*-value/number of contrasts in the five subjects monitored at all time points). Differences were considered statistically significant at *p* < 0.01, while tendencies were reported at *p* < 0.05. Data analysis was performed using Stata v.16.1 (StataCorp LLC, College Station, TX, USA), and graphical analysis were carried out in GraphPad Prism 10.01 (GraphPad Software Inc., La Jolla, CA, USA).

## 3. Results

### 3.1. Impact of BoAHV-1 on Circulating Leukocytes Subsets

The calves exhibited clinical signs from 2 to 10 dpi, as we previously described [18]. All animals survived the infection, except for one calf that died at dpi 4. Necroscopy of the animal revealed pustules in the nasal mucosa, acute haemorrhagic tracheitis with bilateral haemorrhagic pneumonia, and cardiac tamponade caused by hyperacute pericarditis [18].

Circulating levels of leukocytes, divided into granulocytes, lymphocytes, monocytes, eosinophils, and basophils, were evaluated throughout the study (Figure 3). The challenge infection did not significantly alter total leukocyte (white blood cell, WBC) or granulocyte (neutrophils, eosinophils, and basophils) levels at any time point (Figure 3). Only one calf presented a drastic rise in neutrophils values at 2 dpi, reaching 24.5 × 10^6^ neutrophils/mL; this subject died soon after (4 dpi). Despite this peak, differences in neutrophils levels between day 0 and day 2 were not statistically significant (*p* = 0.4375).

*BoAHV-1* infection led to an early decline in circulating lymphocytes levels, with a tendency observed at 2 dpi (*p* = 0.0241). Nevertheless, this alteration was only transient, and lymphocytes returned to pre-infection levels by the end of the experiment (Figure 3). *BoAHV-1* infection also triggered a transient increase in circulating monocyte levels, reaching statistical significance at 4 dpi (*p* = 0.0027), with tendencies observed at 2 dpi (*p* = 0.0322) and 8 dpi (*p* = 0.0206). No significant differences were observed at later time points post-infection (10 and 14 dpi) (Figure 3). The lymphocyte-to-monocyte ratio (LMR) was also calculated, and the data showed that *BoAHV-1* infection caused an early transient reduction in this parameter, with tendencies at 2 dpi (*p* = 0.0260) and 4 dpi (*p* = 0.0222) (Appendix A).

Flow cytometry analysis revealed that infection caused a modest, transient decrease in circulating levels of both αβ and γδ-T cells, with tendencies observed at 2 dpi (*p* = 0.0274 for αβ-T cells, *p* = 0.0354 for γδ-T cells) and 4 dpi (*p* = 0.0483 for αβ-T cells, *p* = 0.0434 for γδ-T cells). No change in the αβ-T cells/γδ-T cells ratio was detected, and both αβ and γδ-T cells values returned to pre-infection levels by the end of the experiment (Figure 4).

We also investigated the distribution of different αβ-T cell subsets, which were classified based on CD4 and CD8 expression into four groups: CD4^+^CD8^+^, CD4^+^CD8^−^, CD4^−^CD8^+^, and CD4^−^CD8^−^. As shown in Figure 5, only a decreasing trend was noted for all subsets at 2 dpi: CD4^+^CD8^+^ (*p* = 0.0312), CD4^−^CD8^+^ (*p* = 0.0453), CD4^+^CD8^+^ (*p* = 0.0312), and CD4^−^CD8^−^ (*p* = 0.0156) (Figure 5).

Regarding γδ-T cells, four different subsets were defined based on the expression of CD8 and WC1: CD8^+^WC1^+^, CD8^+^WC1^−^, CD8^−^WC1^+^, and CD8^−^WC1^−^. As displayed in Figure 6, *BoAHV-1* challenge infection resulted in a transitory decrease of CD8^−^WC1-γδ-T cells at 4 dpi (*p* = 0.0089). A decreasing trend was also observed earlier (2 dpi) in three subsets: CD8^+^WC1^+^, CD8^+^WC1^−^, and CD8^−^WC1^−^ (*p* = 0.0469 for CD8^+^WC1^+^, *p* = 0.0312 for CD8^+^WC1^−^, *p* = 0.014 for CD8^−^WC1^−^) (Figure 6).

Subsequently, changes in circulating B lymphocyte levels were investigated using multicolour flow cytometry. B cells were identified by the expression of CD79 (intracellular marker) and were further differentiated into two subsets based on the surface expression of CD21: CD21^+^ and CD21^−^ (Figure 1). No significant change in circulating B cell levels or their subsets was detected at any time point post-infection, except for an increasing trend in CD21^−^ B cells at 2 dpi (*p* = 0.0183) (Figure 7).

Subsequently, flow cytometry was used to define three main monocyte subsets: classical (cM), intermediate (intM), and non-classical monocytes (ncM). Total monocytes were defined by the expression of CD172a, and then subsets were differentiated based on the expression of the surface markers CD14 and CD16: cM (CD14^++^CD16^−/low^), intM (CD14^+/++^CD16^+^), and ncM (CD14^−/low^CD16^++^) (19). *BoAHV-1* infection triggered a transient increase in cM levels, with statistical significance at 4 dpi (*p* = 0.0206), and tendencies observed at 2 dpi (*p* = 0.0407) and 8 dpi (*p* = 0.0045). A similar increasing trend was also observed for intM levels, with a tendency at 4 dpi (*p* = 0.0232). In contrast, no changes in ncM levels were detected during the study (Figure 8).

### 3.2. Impact of BoAHV-1 on Serum Cytokines Levels

Serum levels of ten key immune cytokines during *BoAHV-1* infection were next assessed. Figure 9 displays the level of three key T cell-associated cytokines: IFN-γ (a Th1 response marker), IL-4 (a Th2 response marker), and IL-10 (a Treg-associated immunosuppressive cytokine) (13). An increasing trend was observed for IFN-γ and IL-10, but not for IL-4. Tendencies were detected in circulating levels of both IFN-γ and IL-10, with a transient early increase observed at 2 dpi (*p* = 0.0218 for IFN-γ, *p* = 0.0359 for IL-10) and 4 dpi (*p* = 0.0229 for IFN-γ, *p* = 0.0123 for IL-10). A small decreasing trend in IL-10 levels was also observed at 14 dpi (*p* = 0.0445) (Figure 9).

In Figure 10, the results of four key pro-inflammatory cytokines are presented: IL-1α, IL-1β, IL-6, and TNF-α [13]. No significant differences were detected in the levels of these pro-inflammatory cytokines, except for a modest decreasing trend in IL-6 at 14 dpi (*p* = 0.0311) (Figure 10). Figure 11 presents the serum levels of two chemokines: MIP-1β and CXCL8. *BoAHV-1* infection induced a transient increase in MIP-1β levels but did not affect CXCL8 concentrations. The rise of MIP-1β circulating values was transient: tendencies were noticed at both 2 (*p* = 0.0312) and 4 dpi (*p* = 0.0250), but they returned to baseline values at later time points (Figure 11). Serum levels of the receptor antagonist IL-36Ra were also examined. *BoAHV-1* infection did not significantly alter IL-36Ra levels, although a minor decreasing trend was noted at 8 dpi (*p* = 0.0328) (Figure 11).

## 4. Discussion

*BoAHV-1* is one of the main causative agents of respiratory and genital diseases in cattle [4,10]. Infection frequently impairs the host’s immune response, contributing to the development of the polymicrobial disorder known as bovine respiratory disease complex (BRDC) [4]. Nevertheless, infected animals often develop a protective immune response, which clears the virus but frequently results in *BoAHV-1* latency, mainly in sensory neurons [4,10].

In this study, the impact of *BoAHV-1* on the bovine immune system was thoroughly analysed using an integrated approach comprising complete blood counts, multicolour flow cytometry, and multiplex ELISA.

Haematological analysis revealed that the infection did not alter the circulating levels of granulocytes. Only one animal showed a marked increase in neutrophil levels at 2 dpi, while no changes in neutrophil, eosinophil, or basophil levels were observed in the other animals throughout the study.

*BoAHV-1* infection resulted in an early and transient decrease in the lymphocyte-to-monocyte ratio. This may be attributed to several factors, including lymphocyte death induced by *BoAHV-1*, redistribution of lymphocytes from the blood to tissues, or enhanced monocyte recruitment in response to infection. With the aim of better understanding our findings, flow cytometry was employed to characterise the phenotypes of lymphocyte and monocyte subsets altered by infection.

A decreasing trend was observed in circulating αβ-T cells (CD3^+^/δ chain^−^) and γδ-T cells (CD3^+^/δ chain^+^), but not in B cells (CD79^+^). Differences among αβ T cell subsets were further investigated using a dedicated multicolour flow cytometry panel. Four αβ-T cell subsets were defined based on the expression of CD4 and CD8. The two main αβ-T cell subsets are CD4^+^/CD8^−^ (T helper; Th) and CD8^+^/CD4^−^ T cells (cytotoxic T lymphocytes, CTL), the former involved in regulating cellular and humoral immune responses, whereas the latter are responsible for killing infected cells [10]. All four subsets (Th, CTL, CD4^+^CD8^+^, CD4^−^CD8^−^) were affected by infection, with a decrease in their levels as early as 2 dpi. Although only tendencies were observed in our work, mainly due to the small number of animals analysed, our finding aligns with previous research showing that *BoAHV-1* can infect CD4^+^ T cells [21], leading to programmed cell death (apoptosis) [21,22,23]. Additionally, other studies have reported that infection with a virulent strain of *BoAHV-1* (Iowa strain) results in a reduction in circulating CD4^+^ and CD8^+^ T cells [24].

The effect of BoAHV-1 infection on γδ-T cells has been investigated in relatively few studies. Amadori et al. (1995) reported an early increase in the prevalence of γδ-T cells among peripheral blood lymphocytes following *BoAHV-1* infection [25], whereas Molina et al. (2013) found no significant changes in circulating γδ-T cell levels after infection with the Iowa strain [24]. A decreasing trend in γδ-T cells was observed in early post-infection; however, the differences were not statistically significant. This might be due to the limited number of tested animals. Consistent with our findings, Risalde et al. (2015) reported a reduction in γδ-T cell numbers in the lungs of infected animals using the same strain [26]. Similarly, Romero-Palomo et al. (2015) reported a reduction of γδ-T cells in the thymus of infected calves [27]. Future studies involving a larger number of animals are needed to more clearly define the impact of *BoAHV-1* on circulating γδ-T cell levels. In our study, we also implemented a specific multicolour flow cytometry panel to distinguish four γδ-T cell subsets based on the expression of Workshop Cluster 1 (WC1) and CD8. WC1 is an extracellular molecule which can act as a pathogen recognition receptor (PRR) and T cell receptor (TCR) co-receptor [28]. WC1^+^ γδ-T cells possess immuno-surveillance functions and have been associated with IFN-γ production, whereas WC1^−^γδ-T cells present mainly a myeloid phenotype [28]. The majority of circulating γδ-T cells are WC1^+^_,_ whereas WC1^−^ γδ-T cells predominate in the gut, mammary gland, and uterus [29]. We observed that *BoAHV-1* infection primarily affected the circulating levels of WC1^−^γδ-T cells, with a longer and more intense decrease compared to WC1^+^ γδ-T cells. This may reflect the redistribution of WC1^−^γδ^−^ T cells from the bloodstream to peripheral tissues during *BoAHV-1* infection. Circulating levels of WC1^+^γδ-T cells were more stable, and previous studies reported that WC1^+^γδ-T cells can release IFN-γ in response to infection [28]. Thus, it is plausible that WC1^+^γδ-T cells contributed to the early increase in circulating IFN-γ levels following infection.

The early alteration of T cell subsets likely underpinned an initial state of immunosuppression, facilitating viral spread within the respiratory tract [4]. Nevertheless, lymphocytopenia was transient; all T cell subsets returned to baseline levels by the end of the experiment, suggesting that a protective immune response developed and ended the loss of circulating immune cells, in agreement with previous observations [15].

Changes in monocyte subsets were also investigated in detail. Monocytes are precursors of tissue macrophages and dendritic cells, and they also play a direct role in inflammation by sensing pathogens, producing cytokines, and presenting antigens to T cells [30]. Haematological analysis revealed that infection promoted a rise in circulating monocyte levels, and flow cytometry was employed to assess differences among subsets. Total monocytes were defined by the expression of CD172a (pan marker for bovine monocytes). Subsets were differentiated based on surface expression of CD14 (LPS co-receptor), and CD16 (FcγIIIR): cM (CD14^++^CD16^−/low^), intM (CD14^+/++^CD16^+^), and ncM (CD14^−/low^CD16^++^) [19]. In cattle, cM constitutes the monocyte subset with the highest phagocytic activity, intM possesses a strong capacity to release pro-inflammatory cytokines and reactive oxygen species (ROS), while ncM are characterised by the lowest phagocytic and ROS-generating capacities [31].

We observed that *BoAHV-1* infection triggered a rise in circulating levels of cM, and an increasing trend was observed also for intM, but not ncM. These data are in line with what we previously observed during Bubaline alphaherpesvirus (*BuAHV-1*) infection in Mediterranean water buffaloes, where a rise of circulating levels of cM and intM, but not ncM, were observed [12]. Given the stronger antimicrobial and pro-inflammatory functions of cM and intM compared to ncM, it is plausible that these subsets are mobilised from the bone marrow during *BoAHV-1* infection to infiltrate infected tissues and counteract virus spread.

Cytokines are small proteins that play a crucial role in coordinating immune responses through complex networks. They have been identified as key immunological and inflammatory biomarkers, and their quantification in biological samples (e.g., laboratory medicine for assessing many infectious diseases [32]. Diverse studies have reported that infection with *BoAHV-1* results in the dysregulation of the cytokine profile of naïve calves [8].

IFN-γ is a type II interferon primarily released by NK cells and activated T cells. It is a hallmark of a Th1 response, which is associated with protection against intracellular pathogens. IFN-γ mediates diverse immune functions, including the enhancement of inflammation, classical activation of macrophages, and stimulation of NK cell activity [33,34]. Infection with virulent *BoAHV-1* was associated with an early increase in serum IFN-γ levels, although only tendencies were detected at 2 and 4 dpi, likely due to the small sample size. This aligns with previous findings [11,12,35]. In addition, other experimental studies with *BoAHV-1* have reported a rise in IFN-γ levels in both infected lungs [26] and nasal secretions [36]. In addition, the presence of IFN-γ-secreting cells was associated with protection against this herpesvirus in previous experiments [37,38,39]. Overall, these studies suggest the host immune system can develop a protective Th1 response against *BoAHV-1*, which clears the infection and halts the loss of circulating T cells and the related immunosuppression status. Future studies should be conducted to characterise the phenotype of IFN-γ-producing cells during infection with *BoAHV-1*.

No major alteration was observed in the circulating levels of IL-4 (pro-Th2 cytokine), in agreement with previous studies [11].

IL-1, IL-6, and TNF-α are important pro-inflammatory cytokines released during the early stages of infection. These cytokines stimulate the release of various chemokines, which in turn enhance the recruitment of leukocytes. Previous studies reported that intranasal administration of a virulent *BoAHV-1* strain (Iowa strain) led to increased levels of IL-1α and TNF-α in the lungs, which were associated with viral clearance [40]. It was also described that infection with the same strain triggered an increase in circulating levels of TNF-α [11,12]. In our study, we did not detect pronounced alterations in the serum values of TNF-α or other pro-inflammatory cytokines (IL-1α, IL-1β, IL-6). It can be speculated that *BoAHV-1* triggered the release of these pro-inflammatory cytokines in the lungs or other infected organs, but this was not reflected in an increase in their serum levels. Alternatively, the limited number of animals included may have masked potential differences. Regarding IL-1β, only one animal presented a drastic rise in IL-1β serum values at 2 dpi, and this subject died soon after showing neutrophilia and severe respiratory clinical signs. It is plausible that neutrophils were the source of IL-1β in this calf, which probably died of a secondary bacterial infection.

This study is the first to investigate the impact of *BoAHV-1* infection on serum levels of the chemokines CXCL8 and MIP-1β. Chemokines are low-molecular-weight mediators that play a crucial role in immune cell recruitment to infected tissues [41]. CXCL8 primarily recruits neutrophils [42], whereas MIP-1β is a potent chemoattractant for monocytes [43]. We observed that wt *BoAHV-1* infection increased MIP-1β, but not CXCL8 serum levels. The observed rise in serum MIP-1β, a chemokine primarily involved in monocyte recruitment, was associated with increased circulating levels of cM and intM early post-infection, suggesting that these cells may contribute to effective viral control and clearance. However, only trends for both MIP-1β and intM were observed, likely due to the limited number of animals tested. Future studies involving larger cohorts should be carried out, comparing also circulating vs. tissue levels of monocyte subsets. Finally, an increasing trend for IL-10 serum levels was observed. IL-10 is an interleukin with strong anti-inflammatory and immunosuppressive activities [44]. In our study, infection led to an early increase in IL-10 serum levels, consistent with previous reports detecting IL-10 in both serum [11] and lung tissue [26]. The early rise of this anti-inflammatory cytokine, concomitant with that of pro-inflammatory MIP-1β, might suggest that a tight control of inflammation is important during *BoAHV-1* infection, to prevent the development of tissue damage. Further studies are needed to investigate the roles of these cytokines during *BoAHV-1* infection, including comparisons between surviving and non-surviving cattle, and between virulent and attenuated viral strains.

## 5. Conclusions

In conclusion, our data revealed that infection with *BoAHV-1* resulted in an initial phase, characterised by a reduction in circulating αβ and γδ-T cell subsets. Subsequently, an increase in circulating monocyte subsets (cM and intM), along with the related chemokine MIP-1β, was observed, suggesting that monocyte recruitment may contribute to the host’s ability to control the infection. Additionally, our findings suggest that IFN-γ and IL-10 play a central role in protection against *BoAHV-1*. These preliminary observations should be investigated in a larger set of animals, analysing also differences between surviving and non-surviving cattle. Overall, the insights from this study improve our understanding of protective immune mechanisms against *BoAHV-1* and may inform the development of more effective vaccines.

## Figures and Tables

**Figure 1 vaccines-13-00996-f001:**
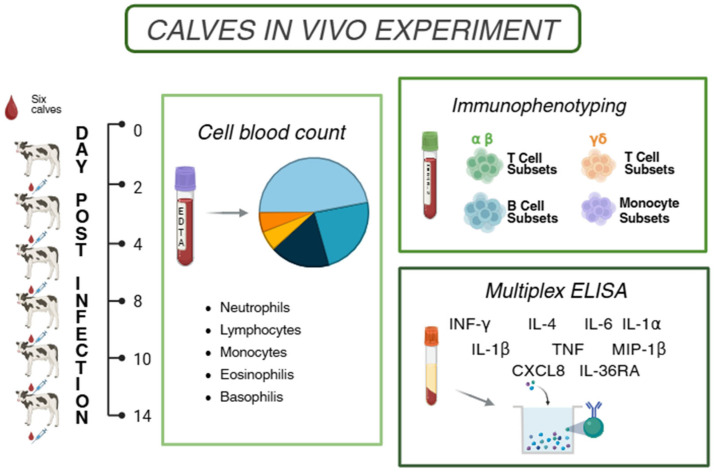
**Study design.** Schematic outline of the experimental study highlighting its key time points. Six healthy Italian Friesian calves were subjected to challenge infection with wt *BoAHV-1* strain 16453/07 TN by the intranasal route. EDTA or Li-Heparin blood and serum samples were collected at 0, 2, 4, 8, 10, and 14 dpi. Figure created with Biorender.com (accessed on 21 July 2025).

**Figure 2 vaccines-13-00996-f002:**
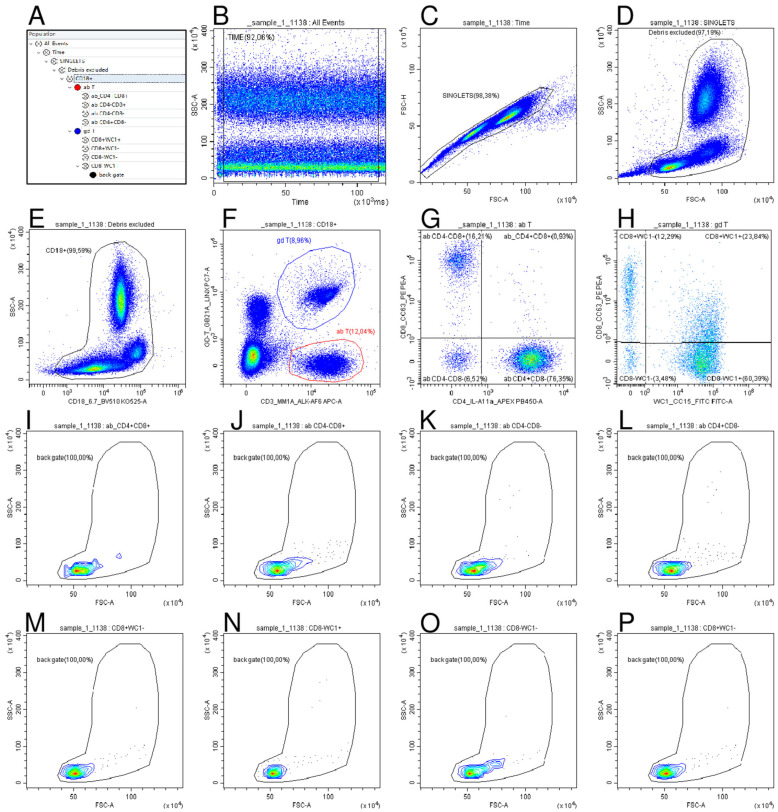
**Gating strategy for T lymphocyte subset identification**. (**A**) Hierarchical representation of the gating strategy used. (**B**) To exclude event bursts, a “TIME” gate was created in the Time vs. SSC dot plot. (**C**) A “SINGLETS” gate was defined in the FSC-A vs. FSC-H dot plot for the exclusion of cellular aggregates. (**D**) A “Debris Excluded” gate was created in the FSC-A vs. SSC-A dot plot for debris exclusion. (**E**) In the CD18 vs. SSC-A dot plot, the “CD18+” gate was set to select all leukocytes, and this gate was then applied to the CD3 vs. GD T dot plot (**F**) to identify αβ T lymphocytes (as CD3+ δ chain−; gate “ab T”) and γδ-T lymphocytes (as CD3+ δ chain+; gate “gd T”), and the relative percentages were used to calculate the absolute counts of αβ and γδ-T lymphocytes shown in Figure 4. (**G**) The “αβ T” gate was applied to the CD4 vs. CD8 dot plot to define the subsets shown in Figure 5, while (**H**) the “gd T” gate was applied to the WC1 vs. CD8 dot plot to characterise the subsets shown in Figure 6. (**I**–**P**). The back-gating analysis confirmed that these final gated populations originated within the expected lymphocyte region in the initial FSC-A vs. SSC-A dot plot, validating the upstream gating strategy.

**Figure 3 vaccines-13-00996-f003:**
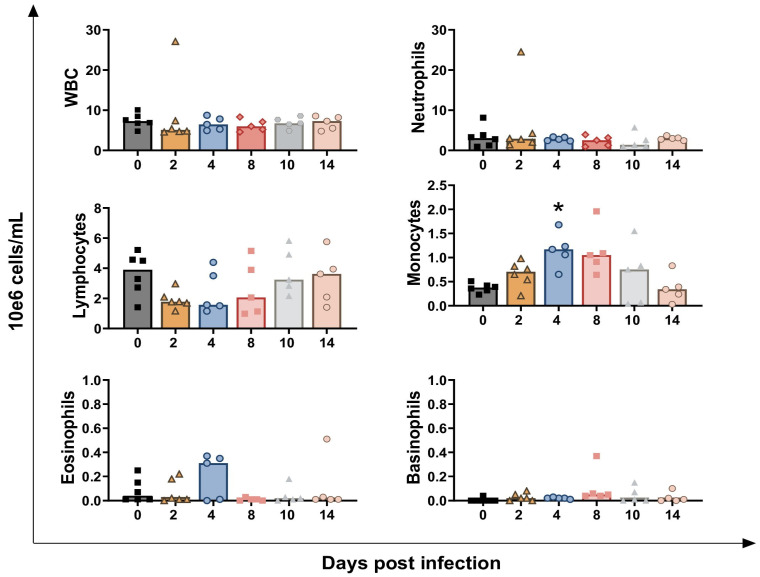
**Circulating leukocytes levels in calves infected with wt *BoAHV-1.*** Six calves were challenge infected with wt *BoAHV-1* by intranasal route. EDTA blood samples were collected before infection (day 0) and post-infection (2, 4, 8, 10, 14 dpi). Changes in the levels of circulating white blood cells (WBC), divided into granulocytes, lymphocytes, monocytes, eosinophils, and basophils, were monitored by a multi-parameter blood cytometric test. For each parameter, values post-infection were compared to those at day 0. * *p*< 0.01.

**Figure 4 vaccines-13-00996-f004:**
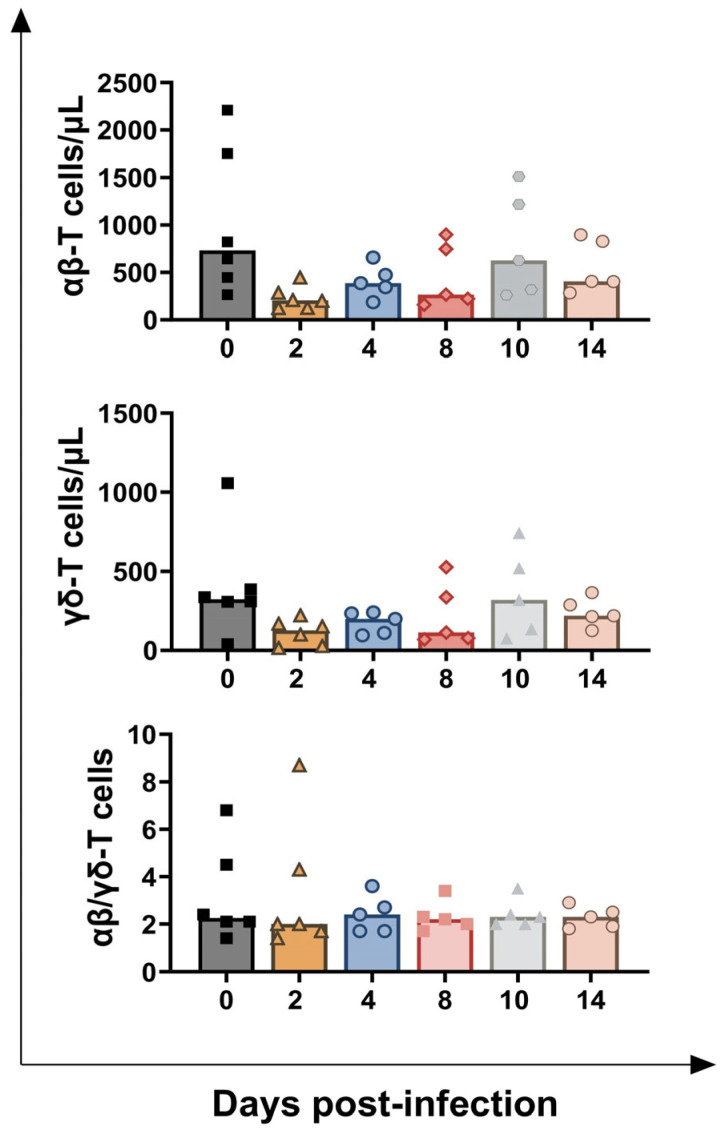
**Circulating levels of T lymphocytes in calves challenge infected with wt *BoAHV-1*.** Six calves were challenge infected with wt *BoAHV-1* by the intranasal route. Heparinised blood samples were collected before infection (day 0) and post-infection (2, 4, 8, 10, 14 dpi). Changes in the levels of T cell subsets (αβ and γδ-T cells) were assessed by flow cytometry. The ratio between αβ-T cells and γδ-T cells was also calculated. For each parameter, values post-infection were compared to those at day 0.

**Figure 5 vaccines-13-00996-f005:**
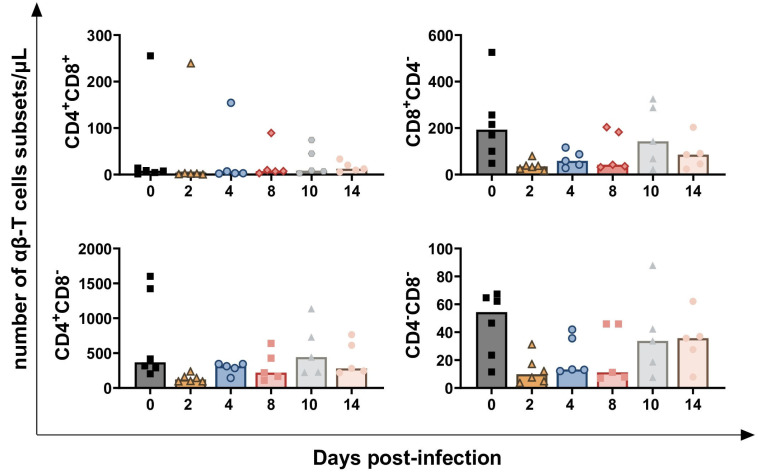
**Circulating levels of αβ-T cell subsets in calves challenge infected with wt *BoAHV-1*.** Six calves were challenge infected with wt *BoAHV-1* by intranasal route, and sera samples were collected longitudinally through the study (0, 2, 4, 8, 10, 14 dpi). Changes in the levels of four αβ-T cell subsets (CD4^+^CD8^+^, CD4^+^CD8^−^, CD4^−^CD8^+^, CD4^−^CD8^−^) were assessed with flow cytometry. For each parameter, values post-infection were compared to those at day 0.

**Figure 6 vaccines-13-00996-f006:**
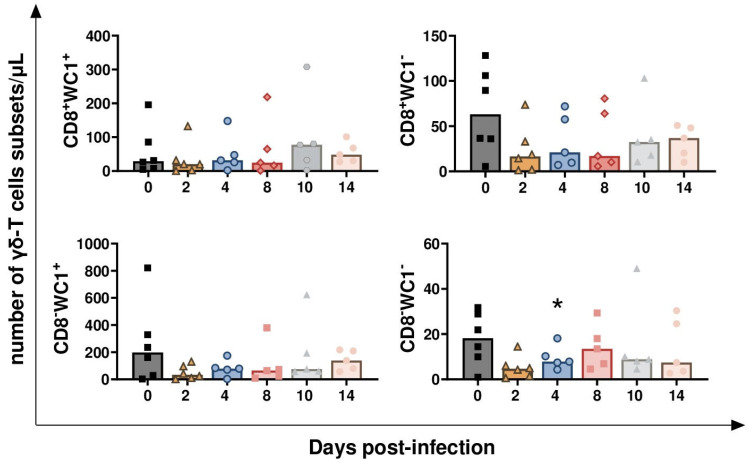
**Circulating levels of γδ-T cell subsets in calves challenge infected with wt *BoAHV-1*.** Six calves were challenge infected with wt *BoAHV-1* by the intranasal route, and blood samples were collected longitudinally throughout the study (0, 2, 4, 8, 10, and 14 dpi). Changes in the levels of four γδ-T cell subsets (CD8^+^WC1^+^, CD8^+^WC1^−^, CD8^−^WC1^+^, CD8^−^WC1^−^) were assessed by flow cytometry. For each parameter, values post-infection were compared to those at day 0; * *p* < 0.01.

**Figure 7 vaccines-13-00996-f007:**
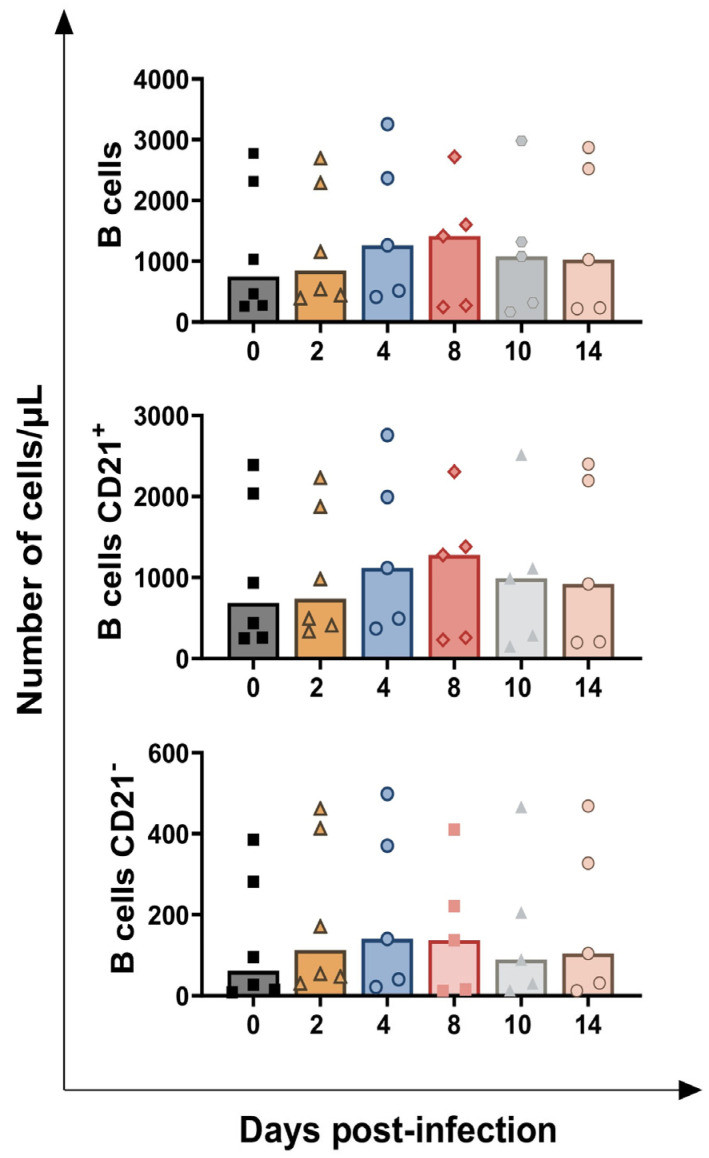
**Circulating levels of B cells and their subsets in calves challenge infected with *BoAHV-1*.** Six calves were challenge infected with wt *BoAHV-1* by the intranasal route. Heparinised blood samples were collected before infection (day 0) and post-infection (2, 4, 8, 10, 14 dpi). Changes in the levels of B cells, then divided into two subsets (CD21^+^, CD21^−^), were assessed by flow cytometry. For each parameter, values post-infection were compared to those at day 0.

**Figure 8 vaccines-13-00996-f008:**
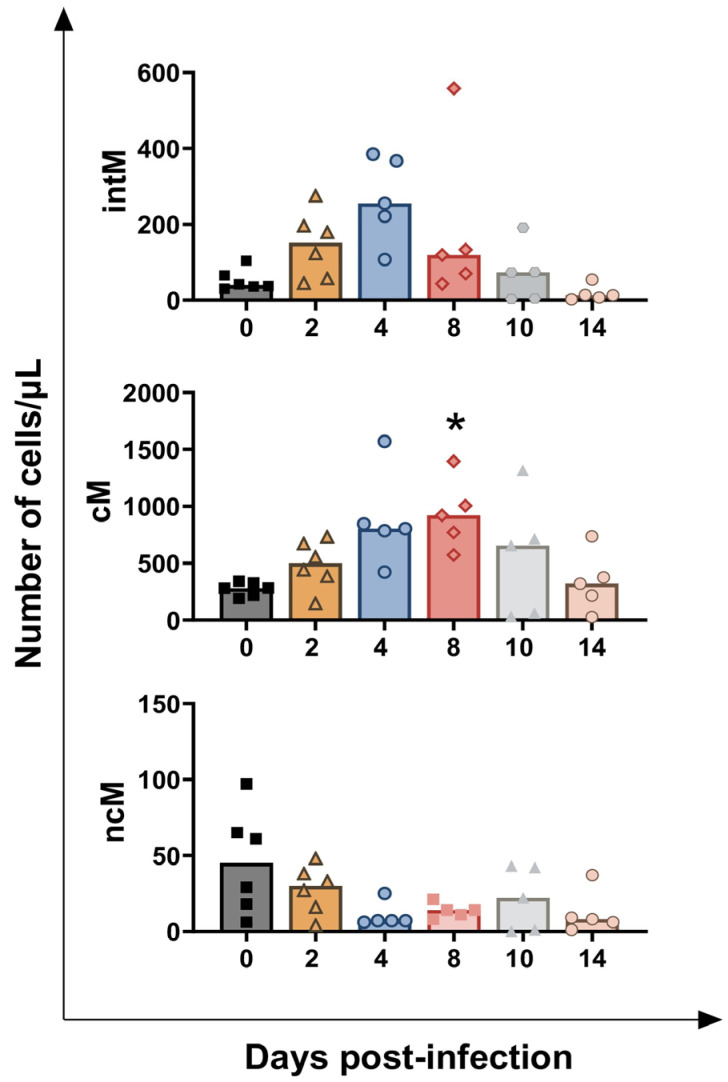
**Circulating levels of monocyte subsets in calves challenge infected with *BoAHV-1*.** Six calves were challenge infected with wt *BoAHV-1* by the intranasal route. Heparinised blood samples were collected before infection (day 0) and post-infection (2, 4, 8, 10, 14 dpi). Changes in the levels of three monocyte subsets (intM, cM, ncM) were assessed by flow cytometry. For each parameter, values post-infection were compared to those at day 0; * *p* < 0.01.

**Figure 9 vaccines-13-00996-f009:**
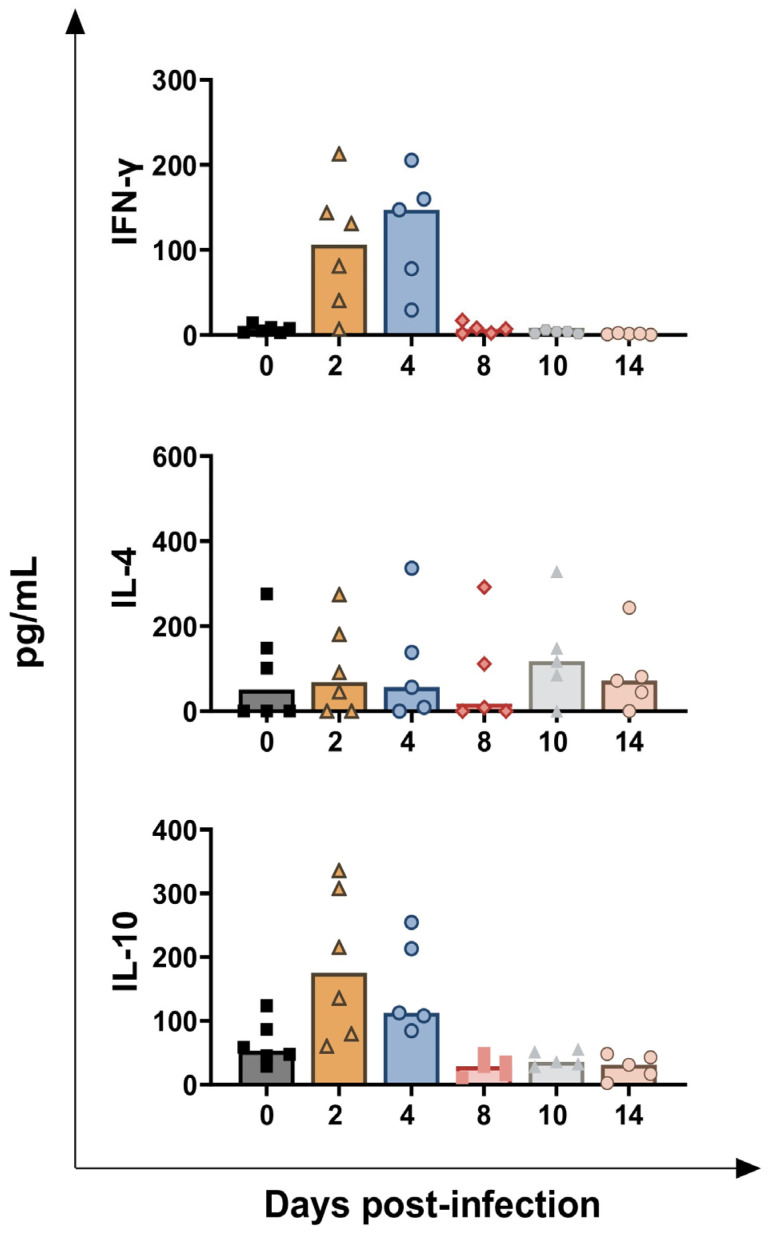
**Circulating levels of T cell cytokines in calves challenge infected with *BoAHV-1*.** Six calves were challenge infected with wt *BoAHV-1* by the intranasal route, and serum samples were collected longitudinally through the study (0, 2, 4, 8, 10, 14 dpi). Changes in the levels of IFN-γ, IL-4, and IL-10 were assessed by ELISA. For each parameter, values post-infection were compared to those at day 0.

**Figure 10 vaccines-13-00996-f010:**
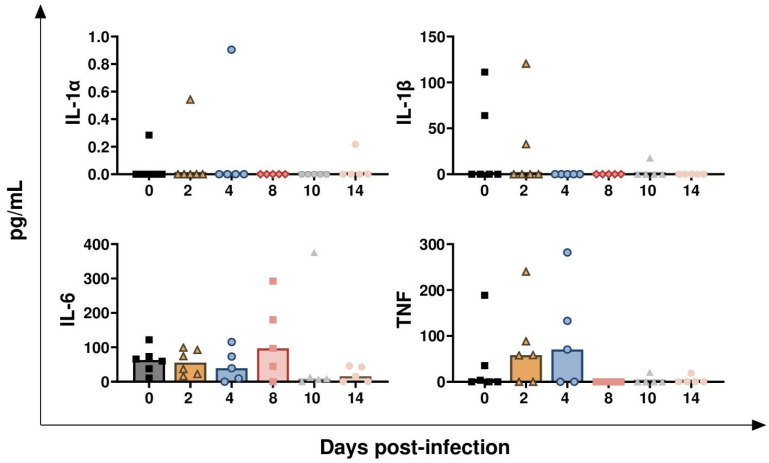
**Circulating levels of pro-inflammatory cytokines in calves challenge infected with *BoAHV-1*.** Six calves were challenge infected with wt *BoAHV-1* by the intranasal route, and serum samples were collected longitudinally through the study (0, 2, 4, 8, 10, 14 dpi). Changes in levels of IL-1α, IL-1β, IL-6, and TNF-α were assessed by ELISA. For each parameter, values post-infection were compared to those at day 0.

**Figure 11 vaccines-13-00996-f011:**
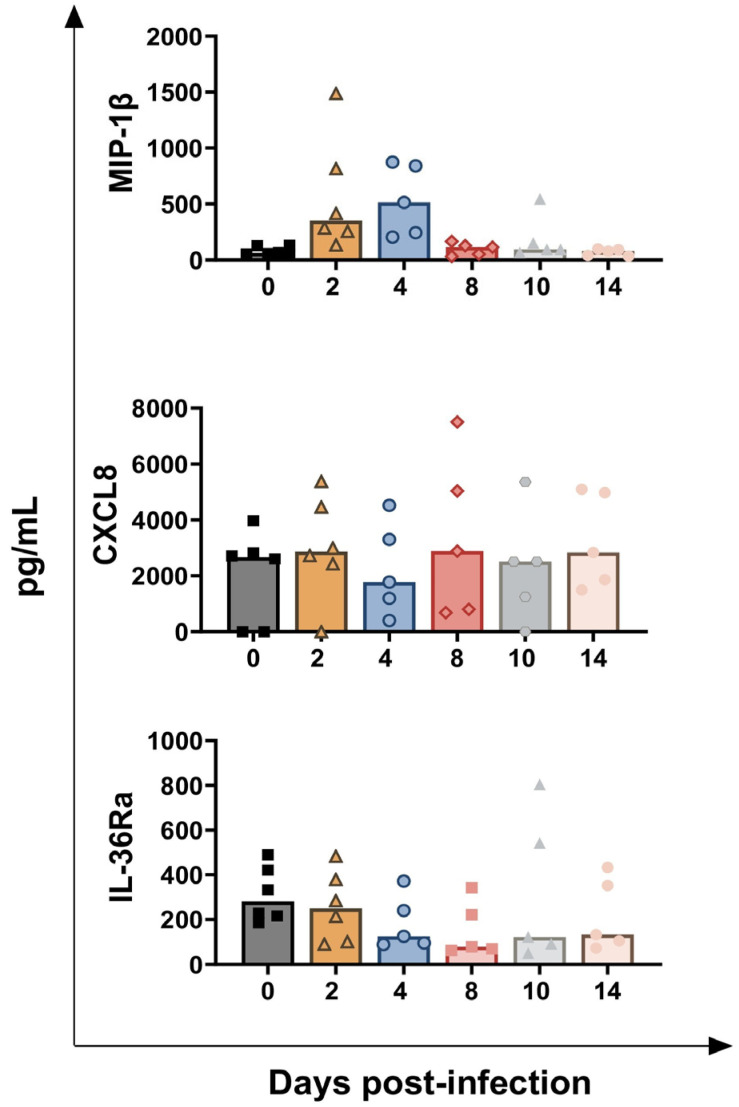
**Circulating levels of MIP-1β, CXCL8, and IL-36Ra in calves challenge infected with *BoAHV-1*.** Six calves were challenge infected with wt *BoAHV-1* by the intranasal route, and serum samples were collected longitudinally through the study (0, 2, 4, 8, 10, and 14 dpi). Changes in the levels of MIP-1β, CXCL8, and IL-36Ra were assessed by ELISA. For each parameter, values post-infection were compared to those at day 0.

**Table 1 vaccines-13-00996-t001:** Monoclonal antibodies used for flow cytometry analysis.

Panel	Antigen	Antibody Clone	Source	Conjugation
Panel 1	CD3	MM1A	WSU-MAC ^1^	Alexa Fluor 647 ^2^
	CD4	IL-A11a	WSU-MAC	Pacific Blue ^3^
	CD8	CC63	Bio-Rad Laboratories	PE
	CD18	6.7	BD	BV510
	δ chain	GB21A	WSU-MAC	PE-Cy7 ^2^
	WC1	CC15	Bio-Rad Laboratories	FITC
Panel 2	CD18	6.7	BD	BV510
	CD21	LT21	Thermo-Fisher	PE
	CD79a	HM47	BD	APC
Panel 3	CD14	TÜK4	Bio-Rad Laboratories	AF750
	CD16	KD1	Bio-Rad Laboratories	FITC
	CD18	6.7	BD	BV510
	CD163	2A10/11	Bio-Rad Laboratories	PE
	CD172a	CC149	Bio-Rad Laboratories	PE-Cy5

^1^ WSU = Washington State University–Monoclonal Antibody Centre, Pullman, WA-USA; ^2^ Purified mAbs were labelled using Lynx Technology (Bio-Rad Laboratories); ^3^ Purified mAb was labelled using Apex Technology (Thermo-Fisher).

## Data Availability

The raw data supporting the conclusions of this article will be made available by the authors on request.

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
