# Peer review of "Characterisation of Cell-Mediated Immunity Against Bovine Alphaherpesvirus 1 (*BoAHV-1*) in Calves"

_vaccines, 2025, doi:10.3390/vaccines13100996_

Round 1
Reviewer 1 Report
Comments and Suggestions for Authors
The article “Characterization of cell-mediated immunity against Bovine alphaherpesvirus1 (BoAHV-1) in calves” reports on a challenge study that collected longitudinal blood samples from experimentally infected calves to assess various immune parameters.
The small sample size and lack of comparison groups limits the impact of the publication. Nonetheless, valuable information may be contained within the report. The major failure is in statistical analysis. Simple pairwise comparisons were done for each analyte examined, always comparing only to D0 baseline values. This precludes evaluating changes that may have occurred between the various time points after challenge. More importantly, performing five pairwise comparisons for each analyte, with an accepted alpha-value of 0.05 for each comparison, increases the likelihood of a type 1 error for at least one of those comparisons to ~20%. This is compounded by evaluating a relative large number of analytes. The statistical methodology needs to be re-evaluated, using either more advanced techniques that evaluate repeated measures, or at the very least, doing a Bonferroni correction for alpha values.
The discussion and/ or conclusion should be expanded to include acknowledgement of limitations of the study and questions that should be addressed in future research. For example, the statement is made “monocyte recruitment and Th1 response seem associated with an animal’s ability to overcome infection.” Additional work is needed to bear this out, including examination of surviving vs. non-surviving cattle, and/ or studies that compare immune response in calves challenged with wild-type and attenuated strains.
Specific comments:
Line 60: “Infected countries” should be “endemic countries…”
Line 61: “Killed vaccine” is a rather dated classification term, with most experts advocating use of the more accurate term “inactivated vaccine.”
Citations do not appear to be correct. Specifically, it is stated that citation 10 reviews failure to utilize an updated immunological toolbox (line 76), and also that citation 10 states circulating levels of IFN-gamma is a clear marker of a Th1 response. Yet the citation list shows 10 to be an original research article examining the interplay of BHV-1 and M. haem. The reviews look to be #13 and 14.
Line 113: Strike “Blood samples were collected longitudinally throughout the study.” This is redundant to the preceding sentence.
Lines 114-117: Explanation of rationale for collection schedule is not appropriate for the materials and methods section. Strike.
Section 2.6: At least five comparisons were made for each analyte (D0 vs. D2; D0 vs. D4; etc.). This is in addition to the large number of analytes being evaluated. As such, it is necessary to do a post-hoc adjustment to reduce the (cumulative) probability of making a (or several) type 1 errors.
Line 287: “…small raise...” would be better stated “…..small rise….”
Lines 301-302: “Differences between subsets were observed. In details” is all extraneous and should be stricken. It is preferable to simply state “BoAHV-1 infection triggered….”
Lines 302, 304, 320, 321, 341, 408, 417, 426, 429, 444, 446, 465, 474, 478, 482, and 490: Raise (or raised) should be rise (or rose)
Lines 315-317: I consider these two sentences unnecessary.
Line 377: Space needed between “subsets” and “cell” (which should be singular—cell, not cells; this occurs frequently through subsequent sentences)
Lines 388-393: You cite four studies as having conflicting results. However, two looked at circulating cells while the other two looked at localized cells (lungs and thymus). I would consider this a notable distinction between the studies and warranting discussion rather than lumping all four into a statement that they are contradictory.
Line 398: Define TCR acronym (not used elsewhere)
Line 466: “…soon later” should be “….soon after”
Line 488: “…including αβ or γδ-T cells” should be AND instead of OR
Reviewer 2 Report
Comments and Suggestions for Authors
The manuscript titled *"Characterization of cell-mediated immunity against Bovine alphaberpesvirus1 (BoAHV-1) in calves"* provides a comprehensive analysis of the immune response to BoAHV-1 infection, integrating hematological, flow cytometric, and cytokine profiling approaches. The study is well-designed and offers valuable insights into the dynamics of leukocyte subsets and cytokine responses during infection. However, several areas require clarification and improvement to enhance the manuscript's impact and reproducibility.
- Figure 3 (Leukocyte Subsets): The transient drop in lymphocytes and rise in monocytes are interesting. Could the authors speculate on the mechanisms driving these changes (e.g., redistribution, apoptosis)?
- Figure 9 (Cytokine Levels): The early rise in IFN-γ and IL-10 is notable. However, the lack of change in IL-4 and pro-inflammatory cytokines (IL-1α, IL-1β, IL-6, TNF-α) is surprising. Could this be due to assay sensitivity or biological factors? A brief discussion would be helpful.
- Figure 2 (Gating Strategy): Include representative plots for all panels (e.g., B-cell and monocyte subsets) in the supplementary materials to ensure transparency.
- Table 1 (Antibodies): Provide catalog numbers for antibodies to enhance reproducibility.
- Abstract: Replace "lymphopenia" with "lymphocytopenia" for consistency with veterinary terminology.
- Line 220: Clarify whether the neutrophil spike in the deceased calf was included in the statistical analysis.
- References: Ensure all citations are formatted consistently (e.g., journal abbreviations in References 8–10).
- The background on BoAHV-1 and its economic impact is well-presented. However, the rationale for focusing on cell-mediated immunity (CMI) over humoral immunity could be strengthened. Are there specific gaps in CMI knowledge that this study addresses?
- The introduction briefly mentions the limitations of current vaccines. It would be helpful to explicitly link these limitations to the study's objectives, such as how understanding CMI could inform vaccine development.
Reviewer 3 Report
Comments and Suggestions for Authors
Manuscript ID: vaccines-3805167
Title: Characterization of cell-mediated immunity against Bovine alphaherpesvirus1 (BoAHV-1) in calves
The article's scientific structure and English are inadequate. Some corrections have been suggested for the text. The entire structure of the article needs to be rewritten. The article's topic has been extensively studied before, so its originality is limited. It was deemed unsuitable for dissemination in its current form.
= English should improve by a native person. The paper suffers from a poor English structure throughout and cannot be published or reviewed properly in the current format. The text contains many typographical, grammatical, and stylistic issues. Please revise and rewrite the entire section for clarity and correctness.
= Abstract: provide the experimental design, statistical analysis statement.
= The novelty of the study needs to be highlighted compared to other similar studies.
= Suggest adding a clear hypothesis or research question in the introduction to guide the reader.
= Although the sample size is adequate, authors should provide power calculation or justification of replication design for statistical robustness.
= A detailed "Conclusion" should be provided to state the final result that the authors have reached. Please note you only need to place your conclusion and not keep putting results, because these have already been presented in the manuscript.
= Line 54: Change " lymphopenia which " to " lymphopenia, which"
= Line 78: Change " which " to "that"
= Line 95: Change "of" to "for"
= Line 127: Change "See" to "see"
= Line 136: Change " In details " to " In detail"
= Line 193: Change " Evaluation of Cytokines Serum Levels " to " Evaluation of Serum Cytokine Levels "
= Line 198: Add "the" before " Bovine "
= Line 222: Change " eosinophils, basophils " to " eosinophils, and basophils "
= Lines 287, 317: Change "were" to "was"
= Line 344: Change " was " to " were "
= Line 377: " though "??
= Line 381: Change " killing of infected cells " to " killing infected cells"
= Line 382: Change " all the four subsets " to " all four subsets "
= Line 464: "of in"??
= Due to the issues identified in this manuscript, I prefer not to proceed with reviewing the other paragraphs.
Round 2
Reviewer 3 Report
Comments and Suggestions for Authors
After the revisions, the quality of the manuscript has seen a significant improvement